# Soil Microfungi of the Colombian Natural Regions

**DOI:** 10.3390/ijerph17228311

**Published:** 2020-11-10

**Authors:** Angela Yaneth Landínez-Torres, Jessika Lucia Becerra Abril, Solveig Tosi, Lidia Nicola

**Affiliations:** 1Faculty of Agricultural & Environmental Sciences, Juan de Castellanos University, 150001 Tunja, Colombia; bioangel19@yahoo.com (A.Y.L.-T.); Jlucia213@gmail.com (J.L.B.A.); 2Department of Earth & Environmental Sciences, University of Pavia, 27100 Pavia, Italy; lidia.nicola01@universitadipavia.it

**Keywords:** fungi, biodiversity, Colombia, Amazonia, andes, caribbean, insular, orinoquía, Pacific

## Abstract

Although Colombia was one of the first tropical countries where an effort was made to gather mycological flora, contributions to the taxonomy, diversity, and ecology of soil microfungi are still scarce. In this study, the diversity of soil microfungi was studied collecting data from literature according to the Colombian natural regions: Andean, Amazonian, Caribbean, Orinoquía, Pacific, and Insular. The majority of the records comes from the Andean region, the most accessible to research. The other regions have been much less studied, with the Insular one with no data at all. International literature reported, up to now, ca. 300 different species of soil microfungi belonging to 126 different genera and 6 phyla (*Ascomycota*, *Basidiomycota Mucoromycota*, *Glomeromycota*, *Mortierellomycota*, and *Olpidiomycota*). Vescicular-Arbuscular fungi were widely investigated with *Acaulospora* and *Glomus*, the most recorded genera with ca. 20 species each. *Ascomycota* was the most diverse phylum with *Penicillium*, *Aspergillus*, and *Fusarium* representing the majority. *Mortierella* is strongly present in *Mortierellomycota*, and in the panorama of all recorded fungi, too. The other phyla and genera were less recorded. It is, therefore, evident the need to continue studying the soil microfungi in Colombia to have a better understanding of soil functioning and its ecosystem services.

## 1. Introduction

Soil hosts an incredible diversity and abundance of microbial life, composed mainly by bacteria and fungi [1]. It is estimated that 1 g of soil contains 10^5^–10^6^ fungal cells [2]. Soil fungi are responsible for a wide array of important ecological functions, such as influencing carbon sequestration through plant life and nutrient mineralization [3]. As tropical zones of Earth are the richest areas in terms of biodiversity and more complex in terms of ecology, their biota is less known than that of the temperate zones, and among the organisms, their fungal component is the one of the least known. One of the first tropical countries where an effort has been made to study the mycological flora is Colombia, beginning in the 20th century [4].

Colombia is located at the extreme north-west of South America. The country is crossed by the Andes mountain range and the Amazon plain, with coasts on the Atlantic and Pacific Oceans. Its continental portion is between 12°26′46″ north latitude and 4°13′30″ south latitude and between 66°50′54″ and 79°02′33″ west longitude, within the fringe Intertropical. Colombian territory covers an area of continental 1,141,748 km^2^ and marine 930,000 km^2^. The country has territorial geographic divisions called “natural regions” differentiated from heterogeneous characteristics of altitude, climate, vegetation, and soil classes, according to these conditions, six regions are distinguished: Amazonia, Andean, Caribbean, Insular, Orinoquía, and Pacific [5,6]. Mycological studies have focused mainly on macroscopic fungi (mainly Basidiomycota) and, on the contrary, contributions to the taxonomy, diversity, and ecology of microfungi are still scarce, and very little is known about the fungi that mainly inhabit the Colombian soil.

The work of Fuhrman and Mayor in 1914 [4] was the first to provide information on Colombian fungi, particularly parasitic ones, and the publication of Chardon and Toro in 1930 [7] was the first to gather mycological literature for a defined political region and presented data from large exploration ranges. Gradually, the characterization of mycological species for Colombia has been enriched through serial publications on the “New or noteworthy fungi from Panama and Colombia”, mainly recorded from the Sierra Nevada of Santa Marta in the administrative region of Magdalena, at 1250–2200 m a.s.l., where some new species were described for the first time, such as: *Haplosporangium lignicum* (decomposing wood), *Dipodascus albidus* (bromelia exudate), *Hyaloria pilacre* (rotting palm stalk), *Hobsonia gigaspora* (bamboo stems and dead palm), *Entonaema liquescens* (Dead wood), *Myxomycidium flavum* (decorticated trunk), *Sclerocystis coccogena* (dead twig), *Tulasnella violea*, and *Cystobasidium sebaceum* [8,9,10,11].

Later, Singer [12] described some mycorrhizogenous fungi on *Quercus* plants, and Dennis [13] presented a brief analysis Colombia’s fungi. In 1971 [14], within a project between the University of Valparaíso (Chile) and University of Pavia (Italy), Rogers verified the adaptive capacity of keratinophilic species to colonize environments near Bogotá, under variable adverse conditions, as well as compared the mycobiota of environments separated by important geographical barriers like the Andes or at different latitudes. In the study, the presence of the following keratinophilic-lithic fungi was reported with their distribution and abundance: *Chrysosporium keratinophilum*, *Trichophyton ajelloi*, *Microsporon gypseum*, *Microsporon fulvum*, and *Trichophyton terrestre.* In 1976, Llanos and Kjøller [15] presented the second part of a research project carried out by the Danish Esso Company in a non-specified area. Soil and associated microorganisms were studied after deposition of oil residues and change in composition community was evaluated, as well as the ability of isolated fungi to attack crude oil and hydrocarbons. The species reported were: *Graphium fructicolum*, *Petriella* sp., *Fusarium oxysporum*, *Penicillium nigricans*, *Paecilomyces lilacinus*, and *Acremonium sclerotigenum*. Guzmán in 1978 [16], presented 95 species of fungi, lichens and myxomycets from Colombia, and discussed their distribution and compared them with Mexican ones. With the “Mycological Flora of Colombia” project, a cooperative program in Mycology was established in 1974, in which more than 2000 collections of fungi were grouped according to the administrative regions of Cundinamarca, Antioquia, Valle, Cauca, and Boyacá [17]. Later, Veerkamp and Gams [18] described new species for science, isolated from agricultural soil and from the Andean forest soil.

The present analysis aims to provide an overview of the current state of knowledge on soil fungal biodiversity of Colombia, in order to establish a starting point for future investigations of the soil–plant–animal–man system, in relation to the pattern of geographical division of the Colombian territory. This study focuses on the variety of soil microfungi, which is a significant source of potential wealth to develop strategies for the rational use and management of available soil resources, both biologically and socially, considering agro-ecosystems and the conservation of biological diversity.

## 2. Materials and Methods

A literature search was performed on Google Scholar and Web of Knowledge using the key words “Colombia” AND “soil” AND “fungi” both in English and in Spanish in order to collect all the publications that represent the state of the art on native soil microfungi in Colombia. The papers and the recorded fungal taxa were divided according to the pattern of geographical division of the Colombian territory that includes six natural regions: Andean, Amazon, Caribbean, Insular, Orinoquía, and Pacific. Only investigations on fungi recorded exclusively from the soil in Colombia were considered and data concerning laboratory or greenhouse experiments, or plant and insect pathogens were excluded. Studies that include the addition of allochthonous microbial inocula into the soil were also excluded. As well, *Uredinales*, although it is the most studied fungal taxon in the country, e.g., [19,20,21,22,23,24], or *Pucciniales*, e.g., [25], have not been considered in the present work, since their main substrate are leaves and stems of plants and many of them are associated to introduced plants. The taxonomic name of the fungi reported in the collected papers were controlled and updated with the current scientific name according to Index Fungorum (http://www.indexfungorum.org/). Taxa were organized at the high-level classification based on Tedersoo et al. [26], and Hibbett et al. [27].

## 3. Results

The papers on soil microfungi of Colombia collected in this work reported a total of 300 identified species belonging to 126 genera. The list of the recorded taxa is reported in Table 1 with their current scientific name. The original taxonomical names are also listed in the table as it was originally reported together with the relative references. The taxa listed were recorded in different areas in Colombian natural regions and the distribution of the sampling sites across the country is portrayed in Figure 1. The most studied region was by far the Andean one, with more than 300 soil fungal taxa, followed by the Orinoquía (47 taxa), Caribbean (38 taxa), Amazon (37 taxa), and Pacific (7 taxa) (Figure 2). The microfungal soil biodiversity of the Insular region was never investigated. The phyla that were more frequently recorded were *Glomeromycota*, *Ascomycota*, and *Mucoromycota* (Figure 3).

### 3.1. Colombian Amazonia

The Amazon region comprises the administrative regions of Guainía, Guaviare, Vaupés, Putumayo, Caquetá, and Amazonas. It includes localities in the administrative region of Vichada, located between 4°10′ south latitude and 4°05′ north latitude and 76°16′ and 66°50′ west longitude, it covers an area of approximately 300,092.7 km^2^ and has elevations between 80 and 500 m. Based on the floristic composition, several types of vegetation have been differentiated: dense forest in the basin of the rivers Caquetá, Amazonas, Putumayo, and Apaporis; dense forests and savannahs of the terraces and erosion surfaces and high hills of the Vaupés river, as well as a mix of forests and savannahs of the Guainía region. There are 5400 species of spermatophytes belonging to 1620 genera and 240 families, whose main use is for folk medicine, food, and timber exploitation, construction of houses, boats, dyes, poisons, and in the manufacture of articles of domestic use. In terms of fauna, there are records of 147 species of reptiles, 868 birds, 95 amphibians, and 210 mammals. The protected area is 48,552.9 km^2^ and has indigenous reserves. The main threats the region is facing are oil prospecting and exploitation (chemical, physical, biological and cultural disturbance), deforestation by logging and burning, as well as the exploitation of wild animal skins [28].

For the Amazon region, 33 fungal species belonging to 16 identified genera and 3 phyla are reported. For the phylum *Glomeromycota* 18 species belong to 10 genera with *Acaulospora*, and *Glomus* the most recorded. For *Glomus*, 4 species were identified and more than 10 non-identified morphotypes were listed. For *Ascomycota*, 14 species were reported, with *Aspergillus*, *Penicillium*, and *Trichoderma*, the most recorded genera. For the phylum *Mucoromycota*, *Gongronella butleri* was reported. Studies on soil fungi for all administrative regions in the Amazon region are reported mainly from natural coenoses. The primary land use considered in the studies was the natural one: Forest (*Inga* spp.) and low intervened forest, the agricultural one (*Eugenia stipitata*), stubble and grassland (*Brachiaria decumbens*). For taxonomic identification the investigations mainly used keys based on morpho- dimensional characteristics. Four studies focused on mycorrhizal fungi. Cardona et al. [29] studied the abundance of arbuscular mycorrhizae in soils under forest cover and pasture in areas of high, medium and low anthropogenic intervention. Subsequently, Peña-Venegas et al. [30] gave an essential contribution to the Illustrated Catalog of Arbuscular Mycorrhizae of the Colombian Amazon based on more than 400 sample analyses. The natural presence of arbuscular mycorrhizae was also evaluated in acid soils of clay-loamy to the clayey texture of Southern Colombian Amazon under forest, stubble, and grassland, at two different depths [31]. Finally, Cardona et al. [32] studied arbuscular mycorrhizal fungi associated with the rhizosphere of chili pepper plants (*Capsicum*) and found 9 morphotypes, 6 of which belonged to the genus *Glomus*, and 3 to the genus *Acaulospora*. The remaining three studies focused on functional solubilizers of phosphate. Vera et al. in two different papers [33,34] evaluated the distribution of 18 fungal isolates in Amazon soils under Arazá (*Eugenia stipitata*) cultivation and evaluated their ability to solubilize phosphate. In the Amazon trapezium, Useche et al. [35], evaluated the abundance and distribution of phosphate solubilizing bacteria and fungi under three soil uses: low-intervening forests, stubble, and pastures at two depths in the soil. Besides, they studied relationships between the abundance of the phosphate solubilizing microorganisms and the physicochemical characteristics of the soil to establish their ecological role in the phosphorus cycling in the soils of the study area.

### 3.2. Colombian Andean Region

The region of the Colombian Andes comprises the administrative regions of Boyacá, Caldas, Cundinamarca, Huila, Norte de Santander, Quindio, Risaralda, Santander, Tolima, and partially the administrative regions of Antioquia, Cauca, Cesar, Chocó, Nariño, and Valle del Cauca. It is collocated between 11°10′ and 0°30′ north latitude and 73°30′ and 77°30′ west longitude. It covers an area of 287,720 km^2^ and reaches elevations up to 5000 m a.s.l. Inside this region, different areas can be differentiated according to the floristic composition: the forests of the piedmont of the Amazon and the Pacific area, the paramos, and the glaciers of the Sierra Nevada del Cocuy. The paramo is a humid alpine inter-tropical ecosystem, characterized by a dominant herbaceous and shrub-like vegetation, usually between 3400 and 5000 m a.s.l. [36]. There are 11,500 species of higher plants belonging to 200 families and 1800 genera. In terms of fauna, there are 974 species of birds, 484 of amphibians, 177 of mammals, and 277 of reptiles. This region has three independent mountain ranges: the western, central, and eastern mountain ranges that exhibit their own climatic, geological, and structural characteristics, and it has indigenous reserves. The main threats the region is facing are the expansion of the agriculture, the overexploitation of natural resources, pollution and the introduction of alien species [37,38,39].

For the Andean region, 16 papers related to the soil fungi were verified. In summary, 265 species were identified belonging to 129 genera and 6 phyla (Table 1). The researches were carried out in the following Colombian administrative regions: Risaralda, Cundinamarca, Boyacá, Antioquia, Huila, Tolima, Caldas, Quindío, Cauca, and Valle del Cauca. No data on soil fungi were reported for the administrative regions of Santander, Norte de Santander, Cesar, Chocó, and Nariño. The main uses of the soil that were taken into account in the researches were agricultural (potato, coffee, flowers, apple and peach), pastureland, resting grassland, and natural (paramo, frailejon plants of *Espeletia barclayana*, *E. killipii*; scrublands of *Calamagrostis effusa*, *Calamagrostis* sp., *Cortaderia selloana*, *Pernettya prostata*, *Buddleja* sp., *Lunipus albus*, *Dentropanax* sp. and forest of *Gynoxys fuliginosa*, *Weinmannia*, *Hypericum*, *Hesperomeles*, *Diplostephium*, land orchids, epiphytes, hemiepiphytes, lithophytes, woodland with *Eucalyptus globulus*, *Pinus* spp., *Acacia* spp., *Vaccinium meridionale*, *Myrtus communis*). For the taxonomic determination morpho dimensional keys and molecular techniques were used, also based on NGS barcode analysis. Many of the studies carried out in this region focused on specific functional groups of fungi: nitrogen solubilizers, phosphorus solubilizers, ligninolytic, keratinophilic, and cellulolytic fungi. For example, Moratto et al. [40] determined, for the Guerrero Paramo (Cundinamarca), the effect of soil use on phosphate solubilizing fungi and nitrogen-fixing bacteria populations under four different conditions of use (potato plantations, cultivated soils, soils at rest, and forest). In this same study area, Bernal et al. [41] counted cultivable microorganisms (bacteria and fungi), cellulolytic microorganisms and endomycorrhizas from forest leaf litter samples, and found 8 endomycorrhizal morphotypes mainly belonging to *Glomus* and *Acaulospora*. Also in the paramo ecosystem, a comparative ecological analysis was carried out [42] based on qualitative and quantitative aspects of the isolated mycoflora, and the major cause of variation in the composition of the fungal communities was related to the vegetation and soil type. Paramo is often an object of mycological analyses and in different papers [42,43], it was highlighted that among soil microfungi *Penicillium*, *Acremonium*, *Cladosporium*, and *Aspergillus* were the most frequently recorded genera in Colombian moorland ecosystems. Avellaneda-Torres and Torres-Rojas [44] characterized the soil bacteria and fungi intp the functional groups of nitrogen fixers, phosphate solubilizers, and cellulolytic in soils under potato crops, livestock, and paramo with little anthropogenic intervention in Risaralda. Moreover, in the paramo ecosystem Álvarez-Yela et al. [45] compared the structure and metabolic functionality of a soil without anthropic intervention with one exposed to agricultural activity (potato cultivation). *Coffea arabica* is one of the most studied plant in Colombia, and, consequently fungal community associated with coffee was deeply investigated. Soil fungal communities were evaluated in coffee plantations in Colombia and Mexico, selecting areas with different intensities of management and edaphic conditions [46]; special attention was paid to iron and calcium phosphate solubilizer fungi. Among the isolated fungi, the presence of *Cylindrocarpon didymum* and *C. obtusisporum* (isolated in Colombia), and *Penicillium janthinellum* and *Paecilomyces marquandii* (isolated in Mexico), was discussed for their potential practical use to improve the phosphate bioavailability [47]. In coffee plantations analyses of vesicular-arbuscular association were also carried out in acidic and phosphate deficient soils under the influence of different physical and chemical parameters [48]. Focusing always on *Coffea*, Bolaños et al. [49] evaluated the diversity and quantity of mycorrhizae associated with the rhizosphere of cultivated plants at the National Coffee Research Center-Cenicafé. Other important plants in Colombia are orchids and its associated mycorrhizal fungi were studied in plant from different habitats [50], following the methodology of counting nuclei in young hyphae cells and sequencing the ITS (internal transcribed spacer) region of nuclear ribosomal genes. Regarding rhizosphere, the cultivable microbial populations of functional importance as a nutrient booster were determined in the rhizosphere of *Espeletia* spp. of two paramos in Antioquia [51]. Beltrán-Pineda [52] isolated and characterized phosphate solubilizers fungi (*Scopuraliopsis* sp. and *Penicillum* sp.) from the rhizosphere of potato crops in paramo soils, in order to obtain fungal strains with biofertilizer potential to mitigate levels of degradation of the soils in these protected areas. Two studies focused, on the other hand, on antagonistic fungi. For the Andean region, the antagonism of some *Trichoderma* isolates was evaluated for the control of pathogens such as *Fusarium oxysporum* and *Rhizoctonia solani* [53]. Moreover, fungi isolated from soils and symphylans were tested to determine their antagonist capacity against symphylans itself [54]. In a recent paper [55] the total soil fungal biodiversity in different agro-ecosystems in Soracá (administrative region of Boyacá) using next-generation sequencing was evaluated. The soil biodiversity was compared in apple and peach orchards, in a resting grassland and in a woodland. The metabarcoding technique allowed the detection of not only cultivable fungi but also the un-cultivable ones. The study listed more than 150 described species with Ascomycota representing the dominant taxon. Basidiomycota resulted dominated by the genus *Sebacina* and *Mortierellomycota* was present with 15 species of *Mortierella*, exclusively recorded, up to now, in Soracá.

### 3.3. Colombian Caribbean

The Caribbean region comprises the administrative regions of Guajira, Magdalena, Atlántico, Bolívar, Sucre, Córdoba, and includes localities of the administrative regions of Antioquia and Cesar. It is located between 7°56′ and 12°25′ north latitude and 77°20′ and 71°08′ west longitude, covers an area of approximately 142,000 km^2^ and has elevations from 0 to 865 m. Flora is represented by communities of mangroves, *Heterostachys ritteriana* and *Philoxerus vermicularis* scrub, cardonal with cactaceae and forests of *Anacardium excelsum* and *Ceiba pentandra*. There are 3429 species corresponding to 1160 genera and 246 families, whose primary use is for folk medicine, firewood, food and construction. For the fauna, 32 species of amphibians, 951 birds, 101 reptiles, 133 arachnids, and 434 hymenoptera are reported. The protected area is 1115.1 km^2^ and includes indigenous populations. The main threats faced by the region are mangrove deforestation, extensive cattle ranching, poor management of garbage, actions of illegal armed groups and drug trafficking, port pollution, sedimentation, and water pollution [56].

For the Caribbean region, 5 papers dealt with soil microfungi recording 30 species belonging to 11 genera and 3 phyla were identified (Table 1). These studies concentrated mainly on the phylum Glomeromycota with the identification of 10 genera, with *Glomus* the most recorded with 9 species. Among the Ascomycota, 4 *Aspergillus* species were identified and strains belonging to *Penicillium*, *Paecilomyces*, and *Humicola* were reported. Among the Mucoromycota, the genus *Rhizopus* was identified.

The studies were conducted almost entirely in the administrative region of Sucre, in the municipalities of Sincé, Sampués, Corozal, and Tolu, while a study was conducted in the administrative region of Antioquia, specifically in Chigorodó and Turbo. No records of soil fungi are reported for the administrative regions of Guajira, Atlántico, Bolívar or Córdoba, or for the localities of Cesar that are part of this region. The main uses of the soil considered were for prairies (*Bothriochloa pertusa*, *Dichanthium aristatium*), agricultural (banana), and natural.

The analysis of mycobiota in the Colombian Caribbean focused on the mycorrhizal component, especially in the last years, due to its importance in tropical soils and its influence on the composition of plant communities, especially in forest plantations and in crops of agronomic importance. In this regard, the incidence of arbuscular mycorrhizal fungi was evaluated comparing natural ecosystems and banana agro-ecosystems, by determining the spore diversity and percentage of association [57]. Results indicated the promising benefit of mycorrhizae inhabiting in the natural ecosystem for the recovery of diversity in the banana agro-ecosystem. Pérez et al. [58] compared the diversity of arbuscular mycorrhizae associated with Colosoana (*Bothriochloa pertusa*) and Angletón (*Dichanthium aristatum*) pastures in a cattle farms. Later, Pérez and Peroza [59] recorded arbuscular mycorrhizal fungi associated with the Angletón grass and characterized their colonization percentage, studying the relationship with the different agrological zones. Moreover, focusing on the metabolic functions of the rhizosphere, fungi present on Colosoana grass rootstocks in cattle farms in the municipality of Sincé were studied [60], considering species of phosphate solubilizers from both rhizosphere and endophytic isolates. In the dry and rainy season, the effect of different types of organic and chemical fertilization on the fungal population of the rhizosphere in the agro-ecosystem of *Bothriochloa pertusa* was also evaluated [61]. Keys based on morphology were mainly used for the taxonomic determinations.

### 3.4. Colombian Orinoquía

The region of the Colombian Orinoquía comprises the administrative regions of Arauca, Casanare, Meta and partially the administrative region of Vichada. It is located between 5° and 2° north latitude and 75° and 67° west longitude, it is 154,193.2 km^2^ long and presents elevations between 80 and 500 m. Regarding the floristic composition, this region is differentiated in several savannah types (dry, wet, and flood savannah) and forests. Spermatophytes are represented by 2047 species belonging to 180 families and 807 genres. Their main use is in the traditional medicine, feeding and industry (timber). Regarding the fauna, 28 species of amphibians, 644 of birds, 119 of reptilians, 65 of arachnids, and 359 of hymenoptera are registered. The protected area is 11,888.8 km^2^ long and presents indigenous reserves. The main threats that this region is facing are the prospection and oil exploitation (chemical, physical, biological, and cultural disturb), the presence of illegal armed groups, the intensive and extensive livestock and the uncontrolled exploitation of the forests [62].

Concerning Orinoquía region, 23 species belonging to 4 genera of the phylum Glomeromycota and 19 genera without specific identification belonging to Ascomycota (14), Mucoromycota (14), and Basidiomycota (1). As for other regions, Glomeromycota was the most investigated, with the identification of 8 species belonging to the genus *Acaulospora*, 10 species to the genus *Glomus*, 2 species to the genus *Entrophospora*, 3 species to the genus *Scutellospora* and 1 is reported for the genus *Gigaspora*.

The studies were carried out in the administrative regions of Meta and Casanare, in the municipalities of Puerto Gaitán, Puerto López, Villavicencio, and Villa Nueva. There were no records of studies on soil fungi for the administrative region of Arauca, nor for the Vichada towns that are part of this region. The main soil use considered in the studies was natural (secondary forest), agricultural (corn, soybean, orange, cassava) and prairies (*Brachiaria* spp., *B. brizantha*, *B. dictioneura*, *Arachis pintoi*, *Desmodium ovalifolum*, *Panicum máximum*, *Paspalum notatum*, and *Trachypogon vestitus*). The researches focused mainly on the study of mycorrhizal fungi. Six vesicular-arbuscular mycorrhizal species new for science were described [63]: *Acaulospora appendicula*, *A. longula*, *A. mellea*, *A. morrowae*, *Glomus manihotis*, and *Entrophospora colombiana*. Later, Dodd et al. [64] deepened the role of arbuscular-vesicle mycorrhizae in infertile soils, and evaluated the effect of phosphate uptake on VAM and the host plants. Serralde and Ramírez [65] studied the populations of arbuscular mycorrhizal fungi associated with two corn varieties in acid soils of the piedmont llanero, during five consecutive years. In the same area, arbuscular mycorrhizal fungi associated with grass and leguminous coverages in oxisols soils were identified, and their colonization capacity was evaluated [66]. Other two studies focused on functional ligninolytic and cellulolytic fungi. Ortiz and Uribe [67] isolated ligninolytic fungi from samples of flooded savannah soil from different agricultural uses and later, they identified isolates of ligninolytic and cellulolytic fungi useful for degrading crop residues and improving soil characteristics in the Eastern Plains [68]. Finally, García et al. [69] studied bulk soil fungi, evaluating the impact of three tillage systems in rotation crops soybean-corn. For the taxonomic determination, all studies mainly used keys based on morphology and in one case, molecular techniques were used.

### 3.5. Colombian Pacific

The Pacific region is composed by the Chocó administrative region and partly by the Valle, Cauca and Nariño administrative regions. It is located between 7°13′ and 1°36′ north latitude and between 77°49′ and 79°01′ west longitude, it covers an area of 131,246 km^2^ approx. and has elevations between 0 and 1,100 m. It exhibits different types of vegetation: mangroves (aquatic communities and marshes, formations of banks or beaches), as well as the vegetation of mainland. There are 5474 plant species belonging to 1406 genera and 271 families, whose main commercial use is timber extraction for export. Other uses are traditional medicine, woodworking, construction and firewood. In terms of fauna, there are 127 species of amphibians, 577 birds, 104 reptiles, 101 arachnids and 649 hymenoptera. Indigenous communities inhabit this area. The main threats faced by the region are gold and platinum extraction, indiscriminate tree cutting, shrimp fishing, permanent agriculture and contamination by wastewater discharges [70].

For the Pacific region, the phylum Glomeromycota was unveiled with different morphotypes of *Glomus* (14) and *Acaulospora* (8). The other strains belonged to the phylum *Ascomycota* with 4 genera, mainly anamorphic, whose species were not identified. Among the Mucoromycota, the genus *Rhizopus* was reported.

The studies were carried out entirely in the Valle del Cauca administrative region. There are no records of studies on soil fungi for the administrative region of Chocó, nor for the localities of Cauca and Nariño that are part of this region. The two studies registered for the Pacific region are relatively recent and were carried out in the same locality, the rural area of Citronela and Zabaletas, in Buenaventura (Valle del Cauca), in agro-ecosystems of chontaduro (*Bactris gasipaes*). Riascos-Ortiz et al. [71] isolated and morphologically characterized fungi associated with the rhizosphere of *Bactris gasipaes* in two different production systems. Afterwards, Molineros et al. [72] evaluated the colonization levels of arbuscular mycorrhizal fungi in the roots of *Bactris gasipaes* and determined the influence of rainfall on the colonization of these fungi.

### 3.6. Interregional Studies

In addition to the studies-mentioned above, three studies carried out their research in more than one region. In particular, a research related to the soil fungi associated with banana cultivation, the third most important one after coffee and flowers in Colombia [73]. In this work 20 records belonging to the genera *Glomus*, *Acaulospora*, *Archaespora*, *Claroideoglomus* and *Kuklospora* (*Glomeromycota*) were reported. The effect of the management system in banana crops (monoculture *vs.* polyculture) was evaluated and the edaphic factors that influence richness and diversity of arbuscular mycorrhizal fungi were determined. This study was carried out in the administrative regions of Cundinamarca, Antioquia, and Magdalena. In the work of Veerkamp and Gams [18] three species new for science were described in the interregional area between Andean and Orinoquía: *Trichoderma inhamatum*, *Rhinocladiella phaeophora*, isolated from agricultural soil samples (corn), at 500 m a.s.l., near Acacias (Meta) plant and *Mortierella ornata* isolated from samples of Andean forest floor, at 3100 m a.s.l., in the Puracé National Park (Cauca–Huila). Another interregional area was that investigated by Sieverding and Howeler [74]: The mountain region of the Cauca Department in South Colombia and the Eastern Plains of Colombia in the Meta department. In this study, mycorrhizal vesicle-arbuscular fungi from cultivation with cassava (*Manihot esculenta*) were analyzed, and their frequency evaluated.

## 4. Discussion

The review of the studies on the microscopic fungi present in the soil of Colombia, showed that the Andean region was the most investigated, since the highest number of species and genera were isolated from this region. The Amazon region stands out in second place, but with a significant difference in the number of species and genera reported compared to the Andean one. The Pacific region, on the other hand, presents the fewer reports. The insular region has never been studied and published data on soil fungi were not found for this region. The relatively high number of records from the Andean region may be due to its extension, eco-geographical and climatic diversity and to the fact that it is the most populated region, which could favor its study and therefore a more in-depth knowledge of its mycological diversity.

Interregional studies on soil microfungi present in Colombian, i.e., those developed in administrative regions that are part of two or more natural regions of the country, include the Andean region (Andean-Caribbean and Andean-Orinoquía), perhaps because to its strategic geographical position, which is in the center of the country, close to all the other regions.

The Amazon was the second region with the highest soil fungal records and unlike the other regions, there are information for each of its administrative regions. The number of data is related to the research efforts that universities, institutions and government made over time in this strategic area of life, one of the most interesting hotspot for diversity both biologic and cultural.

The study of soil microscopic fungi has been growing in recent years; however, the need for mycological studies of the soil in the country is still peremptory since every natural region, with the exception of the Amazon, has administrative regions where there is no information regarding this important area of knowledge.

It is essential to continue and complement these studies on soil microscopic fungi in Colombia, as well as to compile, structure, and systematize the existing information and collections through online databases that could allow greater access and understanding of information on biological diversity and ecological aspects of fungi in the Colombian soil. The already existing Integrated Information System-SiB Colombia is an initiative that aims to provide free access to information on the country’s biological diversity, through the “Catalog of the Biodiversity of Colombia” [75]. The on-line catalogue is highly useful to collect information to fungi, above all on Colombian macrofungi. It is important to implement the system with new data on soil filamentous fungi and yeasts.

The analysis of the literature allowed to show that the methodologies for the taxonomic determination should be enriched, since the morphological characterization through identification keys makes it difficult to unveil unculturable fungi. So, it is necessary to couple this type of analysis with molecular and metagenomic approach, although, it must be considered that this methodology can be expensive for many countries, such as Colombia.

In Colombia, special attention has been paid to *Glomeromycota*, consistently to what is reported from other South American countries [76], due to the important ecological role in plant symbiosis that can be exploited in agro-ecosystems.

The genus *Glomus* is dominant in the endomycorrhizal composition, followed by *Acaulospora* and *Rhizophagus*. However, it is necessary to continue studies on this ecological group in Colombia to assess the actual diversity and distribution, especially in areas of the country where they have not yet been carried out, such as Insular region.

The data obtained in this analysis, according to the reported papers, show that the most found genera in Colombian soils are *Penicillium*, *Glomus*, and *Acaulospora* with ca. 20 species each, *Mortierella* with 15 species, *Aspergillus*, *Fusarium*, *Mucor*, *Rhizophagus*, and *Trichoderma* with ca. 10 species each. The other genera were less recorded in the soil but include important taxa for their beneficial effect on soil ecosystem such as *Aureobasidium*, *Beauveria*, *Clonostachys*, *Dactylaria*, and *Metarhyzium*.

Other taxa are less recorded and analyzed. Very few papers give a panoramic picture of soil mycobiota, e.g., those concerning Andes [45,55]. Colombian mycological research, in this respect is in its infancy, above all, if compared with research carried out by other Central and South America countries, such as Cuba and Brazil [77].

## 5. Conclusions

In this work, most of the records of soil microfungi in Colombia reported in the international literature was collected and analyzed. Knowledge on soil mycobiota results strongly unbalanced towards some areas, e.g., Andean region, and taxon such as *Glomeromycota*.

It isessential to continue to study the soil fungal biodiversity, especially in the administrative regions not yet documented, in order to unveil, preserve, and exploit the richness and the diversity of these microorganisms for the health of the soil. Reaching this objective is urgent, since there are strong anthropogenic threats that continually change the environmental conditions of ecosystems and the consequences of which have not been completely known, yet. This is especially true for the endangered regions, such as the Insular one, so little known and where no data was reported for the soil fungal diversity at all.

In order to complement the information about soil fungi in Colombia, it is also essential to understand the influence of abiotic parameters such as altitude, latitude, and edaphic characteristics, on their physiological activity, especially in the contest of the great ecosystem heterogeneity of the country.

Knowledge about the biological diversity of soil fungi as well as the understanding of their ecology will contribute to optimizing the ecosystem services above all linked to agro-environments, recovery of highly anthropogenic areas and conservation of natural habitat, especially considering the great functional potential of soil fungi such as beneficial ones, i.e., arbuscular mycorrhizae, cellulolytic and lignolytic fungi, antagonists, phosphate, and calcium solubilizers.

Implementing the research for understanding how soil differently used and managed affect mycological diversity and vice versa, is a must since the data obtained so far are ambiguous, and in some cases, contradictory. The knowledge about the potential of utilization of mycobiota to improve soil management is fundamental to answer the challenges which must be faced with to protect the renewable resources on the earth, above all in biodiversity rich countries as Colombia.

## Figures and Tables

**Figure 1 ijerph-17-08311-f001:**
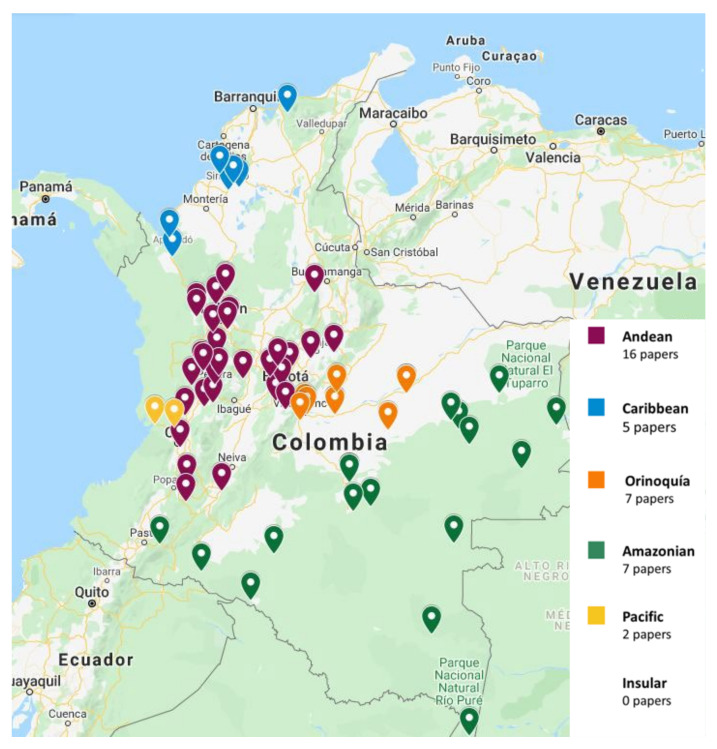
Distribution of the sampling sites of the selected papers across Colombia.

**Figure 2 ijerph-17-08311-f002:**
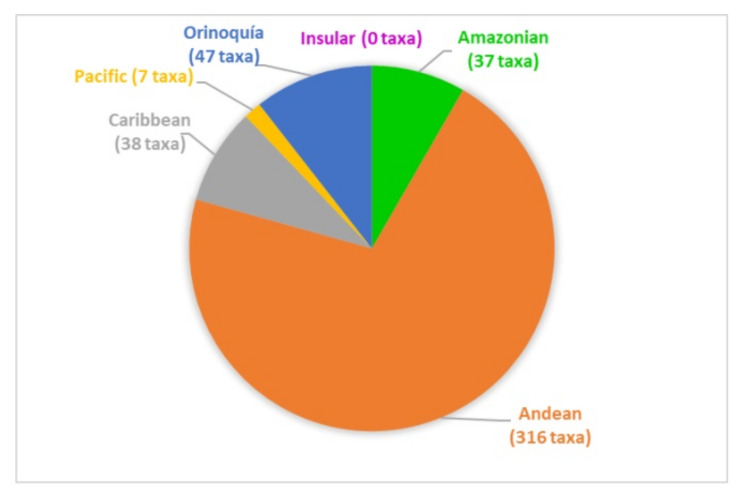
Number of soil microfungal taxa recorded from the natural regions of Colombia.

**Figure 3 ijerph-17-08311-f003:**
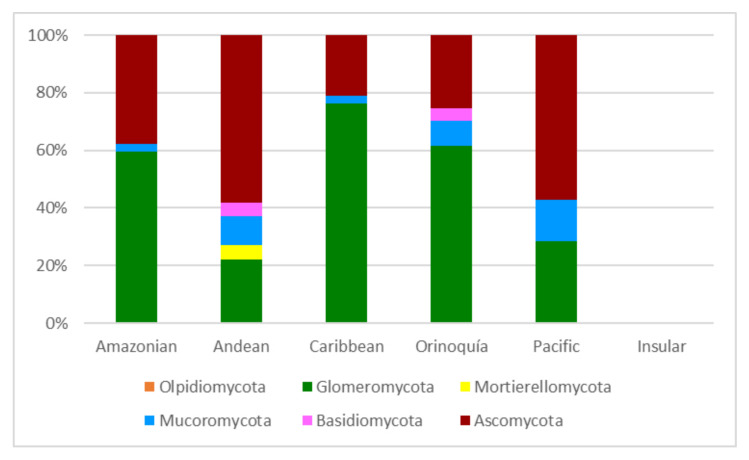
Soil microfungil at phylum level (expressed in %) recorded from the natural regions of Colombia.

**Table 1 ijerph-17-08311-t001:** List of taxa, studying areas in Colombia and references. Regions: Am: Amazon; And: Andean; Car: Caribbean; Ins: Insular; Or: Orinoquía; Pac: Pacific; Int: Interregional areas.

	Colombian Natural Regions and References
Identified Fungal Taxa	Am	And	Car	Ins	Or	Pac	Int
**FUNGI**							
fungi with taxon not specified		[45,51]					
**OLPIDIOMYCETA**							
***Olpidiomycota***							
*Olpidiaster brassicae* (Woronin) Doweld		[55]					
**MUCOROMYCETA**							
***Glomeromycota***							
*Acaulospora alpina* Oehl, Sýkorová and Sieverd.		[48]					
*Acaulospora brasiliensis* (B.T. Goto, L.C. Maia and Oehl) C. Walker, M. Krüger and A. Schüßler							[73]
*Acaulospora capsicula* Błaszk.		[48]					
*Acaulospora cavernata* Błaszk.		[48]					
*Acaulospora excavata* Ingleby and C. Walker							[73]
*Acaulospora scrobiculata* Trappe					[64,66]		[73]
*Acaulospora colombiana* (Spain and N.C. Schenck) Kaonongbua, J.B. Morton and Bever (sub. *Kuklospora colombiana*)		[48]					[73]
*Acaulospora denticulata* Sieverd. and S. Toro		[48]			[66]		
*Acaulospora elegans* Trappe and Gerd.		[48]					
*Acaulospora foveata* Trappe and Janos	[30,31,32]				[64]		
*Acaulospora laevis* Gerd. and Trappe		[48]					
*Acaulospora longula* Spain and N.C. Schenck		[48]			[63,64]		[74]
*Acaulospora mellea* Spain and N.C. Schenck	[30]	[48,49]			[63,64,66]		
*Acaulospora morrowiae* Spain and N.C. Schenck	[30,31]	[48]			[63,64,66]		
*Acaulospora rehmii* Sieverd. and S. Toro	[30]						
*Acaulospora spinosa* C. Walker and Trappe		[48]			[64]		
*Acaulospora splendida* Sieverd., Chaverri and I. Rojas		[48]					
*Acaulospora tuberculata* Janos and Trappe	[30,31]						
*Acaulospora* spp.	[31,32]		[57]		[65,66]	[72]	[73]
*Ambispora appendicula* (Spain, Sieverd. and N.C. Schenck) C. Walker (sub. *Acaulospora appendicula*)		[49]			[63,64]		[74]
*Ambispora fecundispora* (N.C. Schenck and G.S. Sm.) C. Walker (sub. *Glomus fecundisporum*)			[59]				
*Ambispora fennica* C. Walker, Vestberg and A. Schüßler		[48]					
*Ambispora leptoticha* (N.C. Schenck and G.S. Sm.) C. Walker, Vestberg and A. Schüßler		[48]					
*Ambispora leptoticha* (N.C. Schenck and G.S. Sm.) C. Walker, Vestberg and A. Schüßler (sub. *Archaeospora leptoticha*)	[30,31]						
*Ambispora leptoticha* (N.C. Schenck and G.S. Sm.) C. Walker, Vestberg and A. Schüßler (sub. *Glomus leptotichum*)			[58,59]				
*Archaeospora myriocarpa* (Spain, Sieverd. and N.C. Schenck) Oehl, G.A. Silva, B.T. Goto and Sieverd. (sub. *Acaulospora myriocarpa)*					[64]		
*Archaeospora schenckii* (Sieverd. and S. Toro) C. Walker and A. Schüßler							[73]
*Archaeospora schenckii* (Sieverd. and S. Toro) C. Walker and A. Schüßler (sub. *Intraspora schenckii*)		[48]					
*Archaeospora trappei* (R.N. Ames and Linderman) J.B. Morton and D. Redecker		[48]					
*Cetraspora armeniaca* (Błaszk.) Oehl, F.A. Souza and Sieverd.		[48]					
*Cetraspora gilmorei* (Trappe and Gerd.) Oehl, F.A. Souza and Sieverd.		[48]					
*Cetraspora nodosa* (Błaszk.) Oehl, G.A. Silva, B.T. Goto and Sieverd.		[48]					
*Cetraspora pellucida* (T.H. Nicolson and N.C. Schenck) Oehl, F.A. Souza and Sieverd. (sub. *Scutellospora pellucida)*	[30,31]				[66]		
*Claroideoglomus claroideum* (N.C. Schenck and G.S. Sm.) C. Walker and A. Schüßler		[48]					
*Claroideoglomus claroideum* (N.C. Schenck and G.S. Sm.) C. Walker and A. Schüßler (sub. *Glomus claroideum*)			[59]				
*Claroideoglomus claroideum* (N.C. Schenck and G.S. Sm.) C. Walker and A. Schüßler (sub. *Glomus maculosum*)			[58,59]				
*Claroideoglomus etunicatum* (W.N. Becker and Gerd.) C. Walker and A. Schüßler		[48]					[73]
*Claroideoglomus etunicatum* (W.N. Becker and Gerd.) C. Walker and A. Schüßler (sub. *Glomus etunicatum*)			[58,59]				
*Claroideoglomus walkeri* (Błaszk. and Renker) C. Walker and A. Schüßler (sub. *Albahypha walkeri*)		[48]					
*Dentiscutata cerradensis* Sieverd., F.A. Souza and Oehl		[48]					
*Dentiscutata erythropus* (Koske and C. Walker) C. Walker and D. Redecker (sub. *Quatunica erythropus*)		[48]					
*Dentiscutata heterogama* (T.H. Nicolson and Gerd.) Sieverd., F.A. Souza and Oehl (sub. *Gigaspora heterogama*)							[74]
*Dentiscutata heterogama* (T.H. Nicolson and Gerd.) Sieverd., F.A. Souza and Oehl (sub. *Scutellospora heterogama*)					[64,66]		
*Dentiscutata savannicola* (R.A. Herrera and Ferrer) C. Walker and A. Schüßler (sub. *Fuscutata savannicola*)		[48]					
*Dentiscutata savannicola* (R.A. Herrera and Ferrer) C. Walker and A. Schüßler (sub. *Scutellospora savannicola*)					[64,66]		
*Diversispora aurantia* (Błaszk., Blanke, Renker and Buscot) C. Walker and A. Schüßler		[48]					
*Diversispora trimurales* (Koske and Halvorson) C. Walker and A. Schüßler		[48]					
*Diversispora trimurales* (Koske and Halvorson) C. Walker and A. Schüßler (sub. *Glomus trimulares*)			[59]				
*Dominikia aurea* (Oehl and Sieverd.) Błaszk., Chwat, G.A. Silva and Oehl		[48]					
*Entrophospora infrequens* (I.R. Hall) R.N. Ames and R.W. Schneid.		[48,55]			[66]		
*Entrophpspora* sp.					[65]		
*Funneliformis coronatum* (Giovann.) C. Walker and A. Schüßler		[48]					
*Funneliformis dimorphicus* (Boyetchko and J.P. Tewari) Oehl, G.A. Silva and Sieverd., in Oehl, Silva, Goto and Sieverding (sub. *Glomus dimorphicum*)			[59]				
*Funneliformis fragilistratus* (Skou and I. Jakobsen) C. Walker and A. Schüßler [as ‘*fragilistratum’*] (sub. *Glomus fragilistratum*)			[58,59]				
*Funneliformis geosporus* (T.H. Nicolson and Gerd.) C. Walker and A. Schüßler [as ‘*geosporum’*] (sub. *Glomus geosporum*)			[58,59]		[66]		
*Funneliformis kerguelensis* (Dalpé and Strullu) Oehl, G.A. Silva and Sieverd.		[48]					
*Funneliformis mosseae* (T.H. Nicolson and Gerd.) C. Walker and A. Schüßler		[48]					
*Funneliformis mosseae* (T.H. Nicolson and Gerd.) C. Walker and A. Schüßler (sub. *Glomus mosseae*)		[55]					
*Gigaspora albida* N.C. Schenck and G.S. Sm.			[58,59]				
*Gigaspora gigantea* (T.H. Nicolson and Gerd.) Gerd. and Trappe							
*Gigaspora margarita* W.N. Becker and I.R. Hall		[48]					
*Gigaspora* spp.	[30]				[65]		
*Glomus ambisporum* G.S. Sm. and N.C. Schenck		[48]	[59]				
*Glomus boreale* (Thaxt.) Trappe and Gerd.			[58,59]				
*Glomus brohultii* R.A. Herrera, Ferrer and Sieverd.	[30]	[48]					[73]
*Glomus citrícola* D.Z. Tang and M. Zang			[58]		[66]		
*Glomus flavisporum* (M. Lange and E.M. Lund) Trappe and Gerd.		[48]					
*Glomus glomerulatum* Sieverd.	[30]				[64]		
*Glomus hoi* S.M. Berch and Trappe			[58]				
*Glomus liquidambaris* (C.G. Wu and Z.C. Chen) R.T. Almeida and N.C. Schenck ex Y.J. Yao (sub. *Sclerocystis liquidambaris*)		[48]					
*Glomus macrocarpum* Tul. and C. Tul.		[48,49]	[58]		[66]		
*Glomus microcarpum* Tul. and C. Tul.		[48]					[74]
*Glomus multicaule* Gerd. and B.K. Bakshi			[58,59]				
*Glomus radiatum* (Thaxt.) Trappe and Gerd.			[59]				
*Glomus rubiforme* (Gerd. and Trappe) R.T. Almeida and N.C. Schenck (sub. *Glomus rubiformis*)	[30]						
*Glomus rubiforme* (Gerd. and Trappe) R.T. Almeida and N.C. Schenck (sub. *Sclerocystis rubiformis*)		[48]					
*Glomus rubiforme* (Gerd. and Trappe) R.T. Almeida and N.C. Schenck (sub. *Sclerocystis pachycaulis)*		[48]					
*Glomus sinuosum* (Gerd. and B.K. Bakshi) R.T. Almeida and N.C. Schenck	[30]						
*Glomus sinuosum* (Gerd. and B.K. Bakshi) R.T. Almeida and N.C. Schenck (sub. *Sclerocystis sinuosa)*		[49]					
*Glomus* sp. (sub. *Glomus claroides*)			[58]				
*Glomus sp. (sub. Glomus invermayanum)*			[58]				
*Glomus spinuliferum* Sieverd. and Oehl		[48]					
*Glomus taiwanense* (C.G. Wu and Z.C. Chen) R.T. Almeida and N.C. Schenck ex Y.J. Yao			[58]				
*Glomus taiwanense* (C.G. Wu and Z.C. Chen) R.T. Almeida and N.C. Schenck ex Y.J. Yao (sub. *Sclerocystis taiwanensis*)		[48]					
*Glomus* spp.	[29,30,31,32]	[41]	[57,58,59]		[65,66]	[72]	[73]
*Intraspora schenckii* (Sieverd. and S. Toro) Oehl and Sieverd.		[48]					
*Kuklospora colombiana* (Spain and N.C. Schenck) Oehl and Sieverd. (sub. *Entrophospora colombiana*)	[30]	[49]			[63,64]		[74]
*Kuklospora kentinensis* Oehl and Sieverd.		[48]					
*Oehlia diaphana* (J.B. Morton and C. Walker) Błaszk., Kozłowska, Niezgoda, B.T. Goto and Dalpé (sub. *Glomus diaphanum*)			[58,59]				
*Otospora bareae Palenz.*, N. Ferrol and Oehl		[48]					
*Paraglomus brasilianum* (Spain and J. Miranda) J.B. Morton and D. Redecker		[48]					
*Paraglomus laccatum* (Błaszk.) Renker, Błaszk. and Buscot		[48]					
*Paraglomus occultum* (C. Walker) J.B. Morton and D. Redecker		[48,49]					
*Paraglomus occultum* (C. Walker) J.B. Morton and D. Redecker (sub. *Glomus occultum*)		[49]	[58,59]		[64,66]		[74]
*Redeckera fulva* (Berk. and Broome) C. Walker and A. Schüßler [as ‘*fulvum’*] (sub. *Glomus fulvum*)			[58,59]				
*Rhizoglomus microaggregatum* (Koske, Gemma and P.D. Olexia) Sieverd., G.A. Silva and Oehl (sub. *Glomus microaggregatum*)	[30,32]				[66]		
*Rhizoglomus vesiculiferum* (Thaxt.) Błaszk., Kozłowska, Niezgoda, B.T. Goto and Dalpé (sub. *Funneliformis vesiculifer*)		[48]					
*Rhizophagus aggregatus* (N.C. Schenck and G.S. Sm.) C. Walker (sub. *Glomus aggregatum*)			[58,59]				
*Rhizophagus aggregatus* (N.C. Schenck and G.S. Sm.) C. Walker (sub. *Rhizoglomus aggregatum*)		[48]					
*Rhizophagus clarus* (T.H. Nicolson and N.C. Schenck) C. Walker and A. Schüßler (sub. *Glomus clarum*)			[59]				
*Rhizophagus clarus* (T.H. Nicolson and N.C. Schenck) C. Walker and A. Schüßler (sub. *Rhizoglomus clarum*)		[48]					
*Rhizophagus fasciculatus* (Thaxt.) C. Walker and A. Schüßler (sub. *Glomus fasciculatum*)			[58,59]		[64,66]		[74]
*Rhizophagus fasciculatus* (Thaxt.) C. Walker and A. Schüßler (sub. *Rhizoglomus fasciculatum*)		[48]					
*Rhizophagus intraradices* (N.C. Schenck and G.S. Sm.) C. Walker and A. Schüßler (sub. *Glomus intraradices*)	[30]	[49]					
*Rhizophagus intraradices* (N.C. Schenck and G.S. Sm.) C. Walker and A. Schüßler (sub. *Rhizoglomus intraradices)*		[48]					
*Rhizophagus invermaius* (I.R. Hall) C. Walker (sub. *Glomus invermaium*)		[49]			[66]		
*Rhizophagus manihotis* (R.H. Howeler, Sieverd. and N.C. Schenck) C. Walker and A. Schüßler (sub. *Glomus manihotis*)	[30,31]				[63,64]		[74]
*Rhizophagus prolifer* (Dalpé and Declerck) C. Walker and A. Schüßler [as ‘*proliferus’*] (sub. *Rhizoglomus proliferum*)		[48]					
*Sclerocystis* sp.			[59]				
*Scutellospora alborosea* (Ferrer and R.A. Herrera) C. Walker and F.E. Sanders (sub. *Racocetra alborosea*)		[48]					
*Scutellospora arenicola* Koske and Halvorson		[48]					
*Scutellospora calospora* (T.H. Nicolson and Gerd.) C. Walker and F.E. Sanders		[48]					
*Scutellospora projecturata* Kramad. and C. Walker (sub. *Orbispora projecturata*)		[48]					
*Scutellospora spinosissima* C. Walker and Cuenca	[30]						
*Scutellospora* spp.	[30]	[49]			[64,65,66]		
*Septoglomus deserticola* (Trappe, Bloss and J.A. Menge) G.A. Silva, Oehl and Sieverd.		[48]					
*Septoglomus deserticola* (Trappe, Bloss and J.A. Menge) G.A. Silva, Oehl and Sieverd. (sub. *Glomus desertícola*)					[66]		
*Septoglomus viscosum* (T.H. Nicolson) C. Walker, D. Redecker, Stille and A. Schüßler (sub. *Glomus viscosum*)	[30,31]						
*Septoglomus xanthium* (Błaszk., Blanke, Renker and Buscot) G.A. Silva, Oehl and Sieverd.		[48]					
*Sieverdingia tortuosa* (N.C. Schenck and G.S. Sm.) Błaszk., Niezgoda and B.T. Goto (sub. *Glomus tortuosum*)	[30]						
***Mortierellomycota***							
*Dissophora ornata* (W. Gams) W. Gams, in Gams and Carreiro (sub. *Mortierella ornata*)							[18]
*Mortierella alpina* Peyronel		[42,55]					
*Mortierella calciphila* Wrzosek		[55]					
*Mortierella capitata* Marchal		[55]					
*Mortierella elongata* Linnem.		[55]					
*Mortierella exigua* Linnem.		[55]					
*Mortierella fatshederae* Linnem.		[55]					
*Mortierella gamsii* Milko		[42,55]					
*Mortierella globalpina* W. Gams and Veenb.-Rijks		[55]					
*Mortierella globulifera* O. Rostr.		[55]					
*Mortierella humilis* Linnem. ex W. Gams		[42,55]					
*Mortierella indohii* C.Y. Chien		[55]					
*Mortierella minutissima* Tiegh.		[42]					
*Mortierella sarnyensis* Milko		[55]					
*Mortierella wolfii* B.S. Mehrotra and Baijal		[55]					
*Mortierella zonata* Linnem. ex W. Gams		[55]					
*Mortierella* spp.		[35,40,44]					
***Mucoromycota***							
*Absidia anomala* Hesselt. and J.J. Ellis		[55]					
*Absidia cylindrospora* Hagem		[55]					
*Absidia glauca* Hagem		[55]					
*Absidia repens* Tiegh.		[55]					
*Actinomucor elegans* (Eidam) C.R. Benj. and Hesselt.		[55]					
*Actinomucor* sp.					[68]		
*Backusella lamprospora* (Lendn.) Benny and R.K. Benj.		[55]					
*Circinella simplex* Tiegh.		[42]					
*Circinella* sp.		[40]					
*Cunninghamella echinulata* (Thaxt.) Thaxt. ex Blakeslee		[43]					
*Cunninghamella elegans* Lendn.		[43]					
*Gongronella butleri* (Lendn.) Peyronel and Dal Vesco	[34]						
*Lichtheimia corymbifera* (Cohn) Vuill. (sub. *Absidia corymbifera*)		[43]					
*Mucor abundans* Povah		[55]					
*Mucor circinelloides* Tiegh.		[42,55]					
*Mucor gigasporus* G.Q. Chen and R.Y. Zheng (sub. *Mucor gigaspora*)		[55]					
*Mucor griseocyanus* Hagem (sub. *Mucor circinelloides* var. *griseocyanus*)		[55]					
*Mucor hiemalis* Wehmer		[43,55]					
*Mucor luteus* Linnem. ex Wrzosek (sub. *Mucor hiemalis* f. *luteus)*		[42]					
*Mucor moelleri* (Vuill.) Lendn.		[55]					
*Mucor racemosus* Fresen. (sub. *Mucor racemosus* f. *sphaerosporus)*		[42]					
*Mucor zychae* Baijal and B.S. Mehrotra (sub. *Mucor zychae* var. *linnemanniae*)		[55]					
*Mucor* spp.		[40,44]			[67,68,69]		
*Rhizopus arrhizus* A. Fisch (sub. *Rhizopus oryzae*)		[43]					
*Rhizopus microsporus* Tiegh.		[55]					
*Rhizopus microsporus* Tiegh. (sub. *Rhizopus oligosporus*)		[43]					
*Rhizopus stolonifer* (Ehrenb.) Vuill.		[55]					
*Rhizopus* spp.			[61]		[69]	[71]	
*Syncephalastrum racemosum* Cohn ex J. Schröt.		[43]					
*Umbelopsis autotrophica* (E.H. Evans) W. Gams (sub. *Mortierella ramanniana* var. *autotrophica)*		[42]					
*Umbelopsis ramanniana* (Möller) W. Gams		[55]					
*Umbelopsis vinacea* (Dixon-Stew.) Arx (sub. *Mortierella vinacea*)		[42]					
*Umbellopsis* sp.		[44]					
*Zygorhynchus* sp.		[40,42]			[67,68]		
**DICARYA**							
***Basidiomycota***							
*Ceratobasidium* sp.		[50]					
*Erythrobasidium hasegawianum* Hamam., Sugiy. and Komag.		[55]					
*Filobasidium floriforme* L.S. Olive		[55]					
*Filobasidium magnum* (Lodder and Kreger-van Rij) Xin Zhan Liu, F.Y. Bai, M. Groenew. and Boekhout		[55]					
*Filobasidium stepposum* (Golubev and J.P. Samp.) Xin Zhan Liu, F.Y. Bai, M. Groenew. and Boekhout (sub. *Cryptococcus stepposus*)		[55]					
*Hannaella oryzae* (Nakase and M. Suzuki) F.Y. Bai and Q.M. Wang		[55]					
*Moniliella* sp.		[35,43]					
*Naganishia diffluens* (Zach) Xin Zhan Liu, F.Y. Bai, M. Groenew. and Boekhout		[55]					
*Rhizoctonia* sp.					[69]		
*Rhodotorula graminis* Di Menna		[55]					
*Saitozyma podzolica* (Babeva and Reshetova) Xin Zhan Liu, F.Y. Bai, M. Groenew. and Boekhout		[55]					
*Solicoccozyma aeria* (Saito) Yurkov		[55]					
*Solicoccozyma terrea* (Di Menna) Yurkov (sub. *Solicoccozyma terreus*)		[55]					
*Trichosporon beigelii* (Küchenm. and Rabenh.) Vuill.		[42]					
*Trichosporon* sp.		[44]					
*Trichosporonoides* sp.					[68]		
*Vishniacozyma victoriae* (M.J. Montes, Belloch, Galiana, M.D. García, C. Andrés, S. Ferrer, Torr.-Rodr. and J. Guinea) Xin Zhan Liu, F.Y. Bai, M. Groenew. and Boekhout (sub. *Cryptococcus victoriae*)		[55]					
***Ascomycota***							
*Acremonium chrysogenum* (Thirum. and Sukapure) W. Gams		[55]					
*Acremonium persicinum* (Nicot) W. Gams		[55]					
*Acremonium psammosporum* W. Gams		[55]					
*Acremonium sclerotigenum* (Moreau and R. Moreau ex Valenta) W. Gams		[55]					
*Acremonium* spp.		[40,43]					
*Akanthomyces lecanii* (Zimm.) Spatafora, Kepler and B. Shrestha (sub. *Verticillium lecanii*)		[42]					
*Alternaria alternata* (Fr.) Keissl.		[43]					
*Alternaria infectoria* E.G. Simmons		[55]					
*Alternaria* sp.		[40]					
*Apodus oryzae* Carolis and Arx		[55]					
*Ascochyta medicaginicola* Qian Chen and L. Cai		[55]					
*Ascochyta medicaginicola* Qian Chen and L. Cai (sub. *Phoma medicaginis*)		[42]					
*Aspergillus aculeatus* Iizuka	[34]						
*Aspergillus candidus* Link			[60]				
*Aspergillus flavus* Link		[43]	[60]				
*Aspergillus flavus* var. *oryzae* (Ahlb.) Kurtzman, M.J. Smiley, Robnett and Wicklow (sub. *Aspergillus oryzae*)	[34]	[43]					
*Aspergillus fumigatus* Fresen.	[35]	[43]					
*Aspergillus neoniveus* Samson, S.W. Peterson, Frisvad and Varga (sub. *Emericella nivea*)		[43]					
*Aspergillus nidulans* (Eidam) G. Winter		[43]					
*Aspergillus niger* Tiegh.	[35]	[43]	[60]				
*Aspergillus ochraceus* K. Wilh (sub. *Aspergillus alutaceus*)		[43]					
*Aspergillus proliferans* G. Sm.		[55]					
*Aspergillus rugulosus* Thom and Raper (sub. *Emericella rugulosa*)		[42]					
*Aspergillus terreus* Thom		[43]	[60]				
*Aspergillus* sp. (sub. *A. flavoclavatus*)	[34]						
*Aspergillus* spp.		[44,46,47,54]	[61]		[69]		
*Aureobasidium pullulans* (de Bary and Löwenthal) G. Arnaud		[42]					
*Aureobasidium* sp.		[44]					
*Auxarthron umbrinum* (Boud.) G.F. Orr and Plunkett		[55]					
*Beauveria caledonica* Bissett and Widden		[55]					
*Beauveria* sp.		[44]					
*Berkeleyomyces basicola* (Berk. and Broome) W.J. Nel, Z.W. de Beer, T.A. Duong and M.J. Wingf. (sub. *Thielaviopsis basicola*)		[55]					
*Bionectria* sp.		[44]					
*Boeremia exigua* (Desm.) Aveskamp, Gruyter and Verkley		[55]					
*Botryotrichum murorum* (Corda) X. Wei Wang and Samson		[55]					
*Botrytis cinerea* Pers.		[42]					
*Cephalotrichiella penicillata* Crous		[55]					
*Chaetomidium leptoderma* (C. Booth) Greif and Currah		[55]					
*Chaetomium globosum* Kunze ex Fr.		[55]					
*Chaetomium globosum* Kunze ex Fr. (sub. *Chaetomium cochliodes*)		[42]					
*Chaetomium* spp.		[46,47]					
*Chrysosporium keratinophilum* D. Frey ex J.W. Carmich.		[14]					
*Chrysosporium lobatum* Scharapov		[55]					
*Cladophialophora chaetospira* (Grove) Crous and Arzanlou (sub. *Heteroconium chaetospira*)		[42]					
*Cladosporium cladosporioides* (Fresen.) G.A. de Vries		[42,55]					
*Cladosporium fusiforme* Zalar, de Hoog and Gunde-Cim.		[55]					
*Cladosporium* spp.		[43,46,54]			[68]		
*Clonostachys candelabrum* (Bonord.) Schroers		[55]					
*Clonostachys divergens* Schroers		[55]					
*Clonostachys rosea* (Link) Schroers, Samuels, Seifert and W. Gams		[55]					
*Clonostachys rosea* (Link) Schroers, Samuels, Seifert and W. Gams (sub. *Gliocladium catenulatum*)	[34]	[40]					
*Clonostachys rosea* (Link) Schroers, Samuels, Seifert and W. Gams (sub. *Gliocladium roseum*)		[40,42,43]					
*Coniothyrium* sp.		[42,44]					
*Curvularia brachyspora* Boedijn		[42]					
*Curvularia spicifera* (Bainier) Boedijn		[43]					
*Curvularia* sp.					[69]		
*Cylindrocarpon didymum* (Harting) Wollenw.		[47]					
*Cylindrocarpon* spp.		[40,46,47]					
*Dactylaria fusiformis* Shearer and J.L. Crane		[42]					
*Dactylonectria macrodidyma* (Halleen, Schroers and Crous) L. Lombard and Crous		[55]					
*Dendrodochium* sp.		[54]					
*Dendryphion nanum* (Nees) S. Hughes		[55]					
*Diaporthe columnaris* (D.F. Farr and Castl.) Udayanga and Castl. (sub. *Phomopsis columnaris*)		[55]					
*Diplodia* sp.		[42]					
*Diplogelasinospora* sp.		[44]					
*Dipodascus geotrichum* (E.E. Butler and L.J. Petersen) Arx (sub. *Geotrichum candidum*)		[42,43]					
*Drechslera* sp.		[44]					
*Epicoccum nigrum* Link		[55]					
*Epicoccum nigrum* Link (sub. *Epicoccum purpurascens*)		[42,43]					
*Eupenicillium shearii* Stolk and D.B. Scott		[42]					
*Furcasterigmium furcatum* (C. Moreau and Moreau ex W. Gams) Giraldo López and Crous (sub. *Acremonium furcatum)*		[55]					
*Fusarium avenaceum* (Fr.) Sacc.		[42]					
*Fusarium chlamydosporum* Wollenw. and Reinking (sub. *Fusarium sporotrichioides*)		[42]					
*Fusarium culmorum* (Wm.G. Sm.) Sacc.		[55]					
*Fusarium equiseti* (Corda) Sacc.		[42,43,55]					
*Fusarium graminearum* Schwabe		[42]					
*Fusarium oxysporum* Schltdl.	[34]	[42,43,55]					
*Fusarium poae* (Peck) Wollenw.		[42]					
*Fusarium redolens* Wollenw.	[34]	[42]					
*Fusarium verticillioides* (Sacc.) Nirenberg (sub. *Fusarium moniliforme*)		[43]					
*Fusarium* spp.		[42,44,46,47,54]			[67,68,69]	[71]	
*Fusicolla merismoides* (Corda) Gräfenhan, Seifert and Schroers, in Gräfenhan, Schroers, Nirenberg and Seifert (sub. *Fusarium merismoides*)		[42]					
*Gelasinospora retispora* Cain		[42]					
*Geomyces* sp.		[44]					
*Geotrichum* sp.					[68]		
*Gilmaniella humicola* G.L. Barron		[40,42]					
*Gilmaniella* sp.		[40]					
*Gliocladium* sp.					[69]		
*Gliomastix cerealis* (P. Karst.) C.H. Dickinson (sub. *Acremonium cereale)*		[42]					
*Humicola fuscoatra* Traaen		[42]					
*Humicola udagawae* (Sergeeva ex Udagawa) X. Wei Wang and Houbraken (sub. *Chaetomium udagawae*)		[55]					
*Humicola* sp.		[46]	[61]				
*Hypocrea* sp.		[44]					
*Ilyonectria destructans* (Zinssm.) Rossman, L. Lombard and Crous (sub. *Cylindrocarpon destructans*)		[42]					
*Kernia nitida* (Sacc.) Nieuwl.		[55]					
*Leohumicola levissima* H.D.T. Nguyen and Seifert		[55]					
*Leptosphaeria* sp.		[44]					
*Mariannaea elegans* G. Arnaud		[42]					
*Melanopsammella vermicularioides* (Sacc. and Roum.) Réblová, M.E. Barr and Samuels (sub. *Chaetosphaeria vermicularioides*)		[55]					
*Metacordyceps chlamydosporia* (H.C. Evans) G.H. Sung, J.M. Sung, Hywel-Jones and Spatafora (sub. *Pochonia chlamydosporia*)		[55]					
*Metapochonia bulbillosa* (W. Gams and Malla) Kepler, S.A. Rehner and Humber (sub. *Pochonia bulbillosa*)		[55]					
*Metarhizium marquandii* (Massee) Kepler, S.A. Rehner and Humber		[55]					
*Metarhizium* sp. (sub. *Metarhizium anopsiliae*)		[43]					
*Microascus brevicaulis S.P. Abbott*, *in Abbott*, *Sigler and Currah (sub. Scopulariopsis brevicaulis)*		[43]					
*Microdochium bolleyi* (R. Sprague) de Hoog and Herm.-Nijh.		[55]					
*Microsphaeropsis arundinis* (S. Ahmad) B. Sutton		[55]					
*Microsporum fulvum* Uriburu		[14]					
*Microsporum gypseum* (E. Bodin) Guiart and Grigoraki		[14]					
*Microsporum incurvatum* (Stockdale) P.L. Sun and Y.M. Ju (sub. *Nannizzia incurvata*)		[43]					
*Mycochlamys macrospora* S. Marchand and Cabral		[55]					
*Myrothecium* sp.		[54]					
*Musidium stromaticum* (W. Gams and R.H. Stover) Giraldo López and Crous (*Acremonium stromaticum*)		[55]					
*Neocosmospora solani* (Mart.) L. Lombard and Crous (sub. *Fusarium solani*)		[42]					
*Neonectria obtusispora* (Cooke and Harkn.) Rossman, L. Lombard and Crous (sub. *Cylindrocarpon obtusisporum*)		[47]					
*Neonectria* sp.		[44]					
*Nigrospora oryzae* (Berk. and Broome) Petch		[55]					
*Oidiodendron tenuissimum* (Peck) S. Hughes		[55]					
*Oidiodendron* sp.					[68]		
*Paecilomyces* spp.		[34,35,43,44,46,54]	[60]		[69]		
*Paracremonium inflatum* L. Lombard and Crous		[55]					
*Paramyrothecium roridum* (Tode) L. Lombard and Crous (sub. *Myrothecium roridum*)		[42]					
*Penicillium bilaiae* Chalab. [as ‘bilaji’] (sub. *Penicillium lilacinoechinulatum*)		[55]					
*Penicillium brevicompactum* Dierckx		[43,55]					
*Penicillium canescens* Sopp		[55]					
*Penicillium catenatum* D.B. Scott		[55]					
*Penicillium citreonigrum* Dierckx (sub. *Penicillium cítreo-viridae*)	[35]						
*Penicillium expansum* Link		[43]					
*Penicillium fluviserpens* S.W. Peterson, Jurjević and Frisvad		[55]					
*Penicillium glabrum* (Wehmer) Westling		[40]					
*Penicillium glabrum* (Wehmer) Westling (sub. *Penicillium frequentans*)		[40,42,43]					
*Penicillium implicatum* Biourge	[35]						
*Penicillium italicum* Wehmer		[43]					
*Penicillium janczewskii* K.W. Zaleski (sub. *Penicillium nigricans*)		[42]					
*Penicillium madriti* G. Sm.		[55]					
*Penicillium malmesburiense* Visagie, Houbraken and K. Jacobs		[55]					
*Penicillium roseopurpureum* Dierckx		[55]					
*Penicillium scabrosum* Frisvad, Samson and Stolk		[55]					
*Penicillium simplicissimum* (Oudem.) Thom		[42,55]					
*Penicillium simplicissimum* (Oudem.) Thom (sub. *Penicillium janthinellum*)	[34]						
*Penicillium vancouverense* Houbraken, Frisvad and Samson		[55]					
*Penicillium verrucosum* Dierckx		[42,43]					
*Penicillium vinaceum* J.C. Gilman and E.V. Abbott		[55]					
*Penicillium virgatum* Nirenberg and Kwaśna		[55]					
*Penicillium* spp.		[35,40,43,44,46,52,54]	[60,61]		[67,68,69]	[71]	
*Petriella setifera* (Alf. Schmidt) Curzi		[55]					
*Peyronellae* sp.		[54]					
*Phialocephala humicola* S.C. Jong and E.E. Davis		[55]					
*Phialophora cyclaminis* J.F.H. Beyma		[42]					
*Phialophora* sp.					[68]		
*Phoma leveillei* Boerema and G.J. Bollen		[42]					
*Phoma sp.*		[43]					
*Pleotrichocladium opacum* (Corda) Hern.-Restr., R.F. Castañeda and Gené (sub. *Trichocladium opacum*)		[42]					
*Podospora leporina* (Cain) Cain		[55]					
*Preussia africana* Arenal, Platas and Peláez		[55]					
*Preussia* sp.		[44]					
*Pseudeurotium hygrophilum* (Sogonov, W. Gams, Summerb. and Schroers) Minnis and D.L. Lindner (sub. *Pseudeurotium zonatum)*		[55]					
*Pseudogymnoascus pannorum* (Link) Minnis and D.L. Lindner		[55]					
*Pseudopithomyces chartarum* (Berk. and M.A. Curtis) Jun F. Li, Ariyaw. and K.D. Hyde (sub. *Pithomyces chartarum*)		[43]					
*Pseudopyrenochaeta lycopersici* (R.W. Schneid. and Gerlach) Valenz.-Lopez, Crous, Stchigel, Guarro and Cano (sub. *Pyrenochaeta lycopersici*)		[55]					
*Purpureocillium lilacinum* (Thom) Luangsa-ard, Houbraken, Hywel-Jones and Samson (sub. *Paecilomyces lilacinus*)		[42,43]					
*Pyrenochaetopsis leptospora* (Sacc. and Briard) Gruyter, Aveskamp and Verkley		[55]					
*Rhinocladiella phaeophora* Veerkamp and W. Gams							[18]
*Saccobolus globuliferellus* Seaver		[55]					
*Sarocladium bactrocephalum* (W. Gams) Summerb.		[55]					
*Sarocladium glaucum* (W. Gams) Summerb.		[55]					
*Sarocladium kiliense* (Grütz) Summerb. (sub. *Acremonium kiliense)*		[43]					
*Sarocladium strictum* (W. Gams) Summerb. (sub. *Acremonium strictum*)		[42]					
*Sarocladium subulatum* A. Giraldo, Gené and Guarro		[55]					
*Scedosporium dehoogii* Gilgado, Cano, Gené and Guarro		[55]					
*Scedosporium prolificans* (Hennebert and B.G. Desai) E. Guého and de Hoog (sub. *Lomentospora prolificans*)		[55]					
*Scopulariopsis* spp.		[35,52]					
*Sedecimiella taiwanensis* K.L. Pang, Alias and E.B.G. Jones		[55]					
*Sordaria fimicola* (Roberge ex Desm.) Ces. and De Not.		[42]					
*Stachybotrys chartarum* (Ehrenb.) S. Hughes		[43]					
*Stachylidium bicolor* Link		[55]					
*Stephanonectria keithii* (Berk. and Broome) Schroers and Samuels		[55]					
*Talaromyces ruber* (Stoll) N. Yilmaz, Houbraken, Frisvad and Samson (sub. *Penicillium rubrum*)		[42,43]					
*Talaromyces sp.*		[46]					
*Tetraploa sasicola* (Kaz. Tanaka and K. Hiray.) Kaz. Tanaka and K. Hiray. (sub. *Tetraplosphaeria sasicola*)		[55]					
*Thielaviopsis* sp.						[71]	
*Torula* sp.		[44]					
*Triangularia phialophoroides* (Mouch. and W. Gams) X. Wei Wang and Houbraken (sub. *Cladorrhinum phialophoroides*)		[55]					
*Trichocladium asperum* Harz		[40]					
*Trichocladium canadense* S. Hughes		[40]					
*Trichocladium griseum* (Traaen) X. Wei Wang and Houbraken (sub. *Humicola grisea*)		[40]					
*Trichoderma aureoviride* Rifai	[34]	[53]					
*Trichoderma hamatum* (Bonord.) Bainier		[42,43,53]					
*Trichoderma harzianum* Rifai		[43,53]					
*Trichoderma inhamatum* Veerkamp and W. Gams							[18]
*Trichoderma koningii* Oudem.		[40,42]					
*Trichoderma longibrachiatum* Rifai	[34]						
*Trichoderma virens* (J.H. Mill., Giddens and A.A. Foster) Arx (sub. *Gliocladium virens*)		[42]					
*Trichoderma viride* Pers.	[34]	[42,53]					
*Trichoderma* spp.		[34,40,44]			[68,69]	[71]	
*Trichophyton ajelloi* (Vanbreus.) Ajello		[14]					
*Trichophyton terrestre* Durie and D. Frey		[14]					
*Trichothecium* sp.		[40]					
*Truncatella* sp.		[44]					
*Verticillium* sp.					[67,68]		
*Volutella ciliata* (Alb. and Schwein.) Fr.		[42]					
*Volutella* sp.		[54]

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
