# Peer review of "Soil Microfungi of the Colombian Natural Regions"

_ijerph, 2020, doi:10.3390/ijerph17228311_

Round 1
Reviewer 1 Report
The manuscript is interesting and written well but there are a few corrections that I think they need to do as follow.
- line 46: should be space between citation and it should be [5,6]
- line 51: you need to add reference/s for that paragraph.
- lines 101-102: what you mean of “… or those on plant and insect pathogens were excluded”? I guess you mean “or plant and insect pathogens were excluded”?
- line 107: what is ca.?
- Table1 tooks a lots of space. Need to reduce the space between the lines.
Author Response
First of all the authors thank for the suggestions and corrections of reviewer 1. Our answers are written in italics
- line 46: should be space between citation and it should be [5,6]
The reference numbers were put together in the text as suggested
- line 51: you need to add reference/s for that paragraph.
The authors agree with this suggestion but decide to erase the phrase “The attention to microfungi has increased only in the last decades and its research has been especially directed to metabolites and functional groups in the different ecosystems”, because many of the references are reported later in the text
- lines 101-102: what you mean of “… or those on plant and insect pathogens were excluded”? I guess you mean “or plant and insect pathogens were excluded”?
Yes, Thank you. The sentence was not clear. The correction was inserted and changed in "Only investigations on fungi recorded exclusively from the soil in Colombia were considered and data concerning laboratory or greenhouse experiments, or plant and insect pathogens were excluded"
- line 107: what is ca.?
The word ca. was deleted
- Table1 tooks a lots of space. Need to reduce the space between the lines.
The table 1 was compressed as much as possible and lines were added to help the readers
English was also revised
A version of the ms with highlighted corrections is uploaded

Reviewer 2 Report
This manuscript offer an interesting review of soil fungi biodiversity in Columbia. Collected data have been classified in 6 different areas in Colombian natural regions; Authors summarize the geographical and climatic characteristics of each regions. Authors explained the topics targeted by selected papers: in this way, they justify that this overview isn't exhaustive
The major lack is the catalog effect, which is very difficult to read. Unfortunately this paper is hard and sometimes boring to read. It needs to be more didactic, enriched by data formulated in graph, pie chart, maps ...
Author Response
We want to thank the reviewer 2 for his suggestions and critics. Our answers are in italics
-The major lack is the catalog effect, which is very difficult to read. Unfortunately this paper is hard and sometimes boring to read. It needs to be more didactic, enriched by data formulated in graph, pie chart, maps ...
We agree with the referee. Graphs,and pie charts were added to help the reader. We discussed to the possibility to move the table 1 in supplementary material but decided to propose maintaining it in the main text, because, usually, supplementary materials are not very popular. Table 1 was compressed as much as possible and lines in the table were evidenced to be easier to read. Fig 2 and Fig 3 were added showing the distribution of the papers and phyla recorded.
English was revised
A ms version with corrections highlighted was uploaded

Reviewer 3 Report
Materials and methods need to be developed a little bit.
Discusssion and conclusions:
This section need to be more discussed because results reported by analyses of colombian soil microflora (fungi) did not discussed with others from other countries. There was no references reported in this section.
Conclusion is too mucg general, please provide more details.
Author Response
First of all, we wish to thank the reviewer 3 for the useful suggestions.
Our answers are in italics
-Materials and methods need to be developed a little bit.
Thanks for the suggestion. A part concerning the taxonomic control of the species names was moved from the results to the paragraph of Material and Methods.
-Discusssion and conclusions:
This section need to be more discussed because results reported by analyses of colombian soil microflora (fungi) did not discussed with others from other countries. There was no references reported in this section.
Conclusion is too mucg general, please provide more details.
Some references were added and a brief critical comparison was done in the paragraph.
A ms with the corrections highlighted is uploaded
